# Learning from failure: Simulating pandemic agreement negotiations in a global health classroom

Julia Smith[ID]*, Ellie Gooderham[ID], Julianne Piper

Faculty of Health Sciences, Simon Fraser University, Burnaby, Canada

* jhs6@sfu.ca

## Abstract

Serious games, including simulations, are increasingly used in university teaching, including in medical and humanitarian fields, as well as in political science and international relations. There is less evidence of application in global health pedagogy. This article reports and reflects on the use of a simulation of global pandemic treaty negotiations in a Master of Public Health class on Global Health and International Affairs. Through participant observation and thematic analysis of students' reflective essays we found that the simulation enabled deep learning in line with the assignment objectives (to apply learning from past health crises, engage with key concepts, and experience global health cooperation and challenges), as well as prompted critical reflections on moral dilemmas related to global health cooperation and decolonizing global health. The simulation provided students with an opportunity to engage with wicked problems embedded within global health by drawing on multiple perspectives and approaches. While the students ultimately failed to successfully negotiate a pandemic treaty, it was these failures that provided opportunities for deep learning and critical reflection as they questioned constraints on their underlying motivations and actions. This experience suggests simulations can serve as a particularly apt approach for teaching interdisciplinary approaches to global health as they enable students to apply different sets of knowledge to a particular problem, explore unfamiliar concepts, and critically assess their assumptions.

## Author summary

This article reports on the use of a serious game, developed around the concept of a pandemic treaty negotiation, in a global health graduate class. It finds that while the students 'failed' to negotiate a pandemic treaty, the simulation enabled deep learning in line with the assignment objectives, as well as prompted critical reflections on moral dilemmas related to global health cooperation

**Data availability statement:** The data are not publicly available due to privacy or ethical restrictions as outlined in the ethical approval from the Office of Research Ethics at Simon Fraser University. Queries can be directed to Director, SFU Office of Research Ethics, at dore@sfu.ca.

**Funding:** This work was supported by Health Research BC (through a Micheal Smith Scholar Award to JS). This work was funded in part by the Pandemics and Borders Project which receives funding from the Canadian Institutes of Health Research and New Frontiers in Research Fund administered by Social Sciences and Humanities Research Council of Canada (SSHRC). The funders had no role in study design, data collection and analysis, decision to publish, or preparation of the manuscript.

**Competing interests:** The authors have declared that no competing interests exist.

and decolonizing global health. It suggests that serious games can provide an innovative and effective approach to global health teaching, calling for greater documentation and analysis of pedagogic outcomes.

## Introduction

Serious games, defined as games that aim to promote better human understanding and decision-making with respect to a particular phenomenon, were developed out of the tradition of war games, used by militaries to test new tactics and teams [1]. Serious games – which include simulations– are now used in humanitarian and other sectors to create a safe-to-fail environment within which individuals can actively engage with situations they may encounter in the field. Increasingly, serious games are also used in university teaching, particularly in the fields of political science and international relations (IR). Evidence from these applications document positive pedagogical effects of serious games including the development of communication skills, teamwork, and critical thinking [2–4]. In particular, serious games have proven apt at "bringing complex theory to life" by providing students with the opportunity to experiment and grapple with operationalizing abstract concepts [5,6]. Multiple evaluations emphasise that simulations promote active learning and encourage students' to consider wicked problems from multiple perspectives [7].

This article aims to add to the evidence on serious games in global health pedagogy by reporting on the application of a simulation of negotiations around a global pandemic treaty in a Master of Public Health class on Global Health and International Affairs. We first ask if the simulation met its stated learning objectives and then, through further qualitative analysis of student reflective essays and participant observations, explore additional pedagogic outcomes. We situate this analysis within existing scholarship on the pedagogy of serious games for teaching and learning, before setting out the design and gameplay of the simulation, titled Accord. The results and discussion offer an analysis of student reflections and outcomes of the simulation for interdisciplinary, applied graduate learning.

### Use of serious games in higher education

The most common simulation in university political science or IR courses is the Model United Nations (Model UN) exercise. Students take on the role of member state diplomats and try to achieve global cooperation around specific challenges. Model UN simulations offer opportunities for deep learning, defined broadly as developing the ability to examine new ideas critically, tying them to existing concepts and 'real world examples' while making numerous links between ideas [8]. While requiring the application of theoretical and abstract concepts, such activities are generally exciting and enjoyable to students. With these dynamics in mind, McIntosh notes that it is important for the instructor to ensure the game does not overwhelm the learning, requiring a confident and well planned facilitator [9]. While recognizing the need for detailed preparation, empirical evaluations of Model UN simulations repeatedly document

their success as an experiential and problem based learning approach that develops critical thinking abilities, enhance communication skills, and fosters deep learning around global issues [10].

In addition to Model UN, IR and political science simulations can focus on peacekeeping missions, the processes of other international organizations (such as the North Atlantic Treaty Organizations), or humanitarian relief operations. For example, an environmental negotiation simulation that drew on both scientific and policy knowledge allowed students to explore the complexity of policy negotiations with lawmakers [9]. As a classroom teaching tool, simulations offer alternatives to teacher-centric methods by incorporating problem-based or experiential approaches that enable self-guided learning and the direct application of students' knowledge [11]. Students engaged in intensive simulation exercises can gain deep learning, as they are confronted with moral dilemmas, competing arguments and solutions, and broader questions of what purpose knowledge serves, while also developing skills such as communication. Newmann and Twigg argue that "the simulation format provides students a better framework than do lecture notes for long term retention of important international relations concepts" [12]. Kraine and Shadel find that students who participated in a simulation around world hunger demonstrated a greater degree of knowledge acquisition than students who learned the same material in a traditional classroom format [13]. Haneklaus et el. highlighted how their serious game - on the Sustainable Development Goals of the UN - led to a significant increase in negotiation knowledge following the simulation across reported gender and education levels [14]. Through a simulation of a zombie apocalypse, Horn et al. found that most students improved their comprehension of core IR concepts, and that the greatest improvements were observed among the weakest students [6]. Other evaluations have similarly found simulations can benefit both advanced and novice learners [8], and may even provide more effective learning modalities for students whose education is not be as well facilitated by traditional classroom structures [9,10].

In addition to IR and political science, both humanitarian response and medical training frequently include simulations in their curriculum. These simulations tend to focus on specific incidents and aim at testing students' knowledge, develop their ability to apply learning under pressure, and engage with unknowns. Simulations related to humanitarian relief allow participants to assess their career alignment with the sector, and increase participants' understandings of the experiences of emergency situations and aid workers [15]. Medical training uses simulations to enable students to put learning into practice, improve knowledge and enhance skill development [16,17]. A meta-analysis of serious games in medical education found multiple positive effects including facilitating learners holistic understanding of scientific concepts [18]. The same review reported just two negative results: that in two studies the nature of serious games negatively influenced the relationship between mental workload and learning effect, and that one study reported no significant differences in in-depth learning.

Aiming to go beyond testing knowledge, Bertelsen et al. developed simulated learning exercises for medical students, specifically focused on cross cultural sensitivity [19]. They found that simulations provided students a controlled, safe environment to practice culturally sensitive communication alongside medical skills. Similarly, Mohamed-Ahmed found simulations with medical students were successful in "helping tomorrow's doctors develop insight into global health challenges and may produce more rounded clinicians capable of caring for more culturally diverse populations" [20]. A serious game for nursing and pharmacy students improved trainee attitudes towards people with dementia, particularly in terms of recognition of personhood [21]. In France, a game was developed to support local health authorities their efforts to reduce social inequalities in health through territorial diagnosis [22]. Such cases suggest the potential of serious games to not only be used for knowledge and skill development in health education, but also to promote critical reflection and deep learning as in other disciplines.

The use of simulations in political science/IR, humanitarian relief, and health training suggests pedagogical value to interdisciplinary global health classes, however there are limited examples of this in existing literature. A One Health related simulation in an ungraduated course was found to be successful in meeting course critical thinking learning objectives as well as helping students understand and apply systems thinking [23]. Others have used games to educate

the public about global health threats finding that games may increase the public's interest in global health. Simulations specific to global health and the World Health Organization (WHO) have been more limited, but are growing in popularity. For example, the School of Global Health at York University facilitated an annual simulation of the World Health Assembly (WHA) to "enhance collaborative governance skills in addressing global health challenges" [24]. The American Mock World Health Organization conference, was organized by the University of North Carolina at Chapel Hill, over a number of years to enable students to put learning around health diplomacy into action, with post event surveys finding the simulation "allowed a paradigm shift not possible in class" [25]. Similarly, Kahlid et al. found a World Health Organization (WHO) simulation bolstered students' research capabilities, critical analysis, time management, and organizational effectiveness [26]. Students also reported improvements in effective interpersonal communication, public speaking, networking, and solution-driven dialogues. The instructors concluded that "simulation-based experiential learning stands out as a transformative educational approach... However, a knowledge gap exists in how to effectively develop and implement these simulations" [27]. Loutet et al., who developed a range of global health simulations, similarly noted, "further research is required to assess the long-term educational impact of simulations in global health" [28]. While there is substantial evidence of the benefits of serious games as a pedagogical approach in general, and particularly within the fields of political sciences and medicine, there are few examples or evaluations related to global health pedagogy.

## Methods and materials

### Ethics statement

Ethical approval for this study was granted by Simon Fraser University Institutional Review Board (study # 30002137). Written consent as obtained from all participants.

### The simulation: Accord

The simulation, titled Accord, from which data was collected and analysed, was facilitated for a Global Health and International Affairs course designed to introduce Master of Public Health (MPH) students to scholarship, policymaking, and practice at the intersection of global health and international affairs (full game materials are available from the corresponding author on request). The course learning outcomes included for students to develop familiarity with key concepts and debates in global health governance, scholarship, and practice, and then apply this knowledge to critically analyze selected global health issues and policy responses. The course aimed to integrate decolonizing approaches, including through a syllabus which incorporated diverse scholars and worldviews; an emphasis on student-led learning and activities; and persistent, explicit, and critical engagement with the colonial context within which the university is situated. Different topics were examined throughout the course, including global health institutions, pandemic response, and the political economy of global health. In particular, the class discussed the negotiations processes for revisions to the International Health Regulations and the proposed pandemic agreement, which was being negotiated by WHO member states throughout the duration of the course (September to December 2023).

The simulation was facilitated on the final day of class. Students were grouped into country teams tasked with negotiating a pandemic treaty, following a COVID-like pandemic in the fictional world of Zecan (Fig 1). Using this fictional setting of Zecan, as opposed to existing states, removed pressure to ensure realism and prevented participants from becoming distracted by the nuances of real world events, perceptions, or ideas. Analysis of serious games has found the use of fictional elements increase game enjoyment and enables players to explore plausible alternatives [29,30]. As been found with games like the Zombie apocalypse noted above, fictional settings reduce bias, making it easier for participants to step into unfamiliar roles and undertake creative thinking [31]. The goal was to create a familiar world unencumbered by the histories and other assumptions we bring to 'real world' engagements. Zecan comprises two high-income (HICs),

**Global Public Health**
PLOS

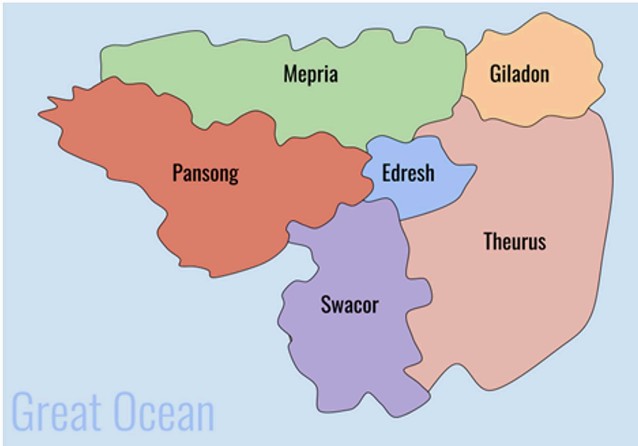

**Fig 1. Fictional world of Zecan.**

two-middle income (MICs), and two low-income (LICs) countries. These differences were operationalised in the simulation through different allocations of Political Resource Points (PRP). PRP represented income and international influence, and initial allocations ranged from two (for LICs) to six (for HICs). These countries also differed in political systems (from parliamentary democracies to one party authoritarian states), and public health capabilities (from strong public systems to under resourced, fragmented systems).

The learning objectives of Accord were to: apply learning from past health crises and global health governance efforts; actively engage with key concepts around global health and international affairs; experience strategies and challenges related to global health cooperation. The simulation was designed as a cooperative game with the overall aim to negotiate a global pandemic treaty that at least five of the six country teams agree to adopt. Individual teams could also earn PRPs throughout the simulation by having articles accepted into the treaty and/or trading them in exchange for support for specific articles. The country that earned the most points by the end of the game would be the 'winner among winners' – but only if negotiations successfully resulted in the adoption of a treaty. If the treaty was not agreed to, all teams would lose, regardless of PRPs earned. This structure aimed to demonstrate both the collective gains of co-operation, as well as the challenges of actualizing them in a diverse context of self-interested actors.

The MPH class involved was small, with just six students, so five additional graduate students/research fellows were recruited to participate in the simulation. All participants were grouped into six teams of two (with two of the authors forming a team) representing the six countries in Zecan. The third author and instructor filled the facilitating role of the Global Health Organization (GHO), which had a similar mandate to WHO. Country teams were provided with background information on the characteristics of each state as well as on their state's priorities for the treaty and relationships with other states. Prior to the simulation, students were instructed to draw on learnings from the course to prepared text for three draft articles they wished to be included in the treaty, based on assigned state contexts. They were also briefed on the final assignment for the course - a reflective essay about the simulation, integrating insights and knowledge from at least seven course syllabus items (i.e., readings, lectures, and student presentations). This prompted students to re-engage with the relevant course materials, such as those on global health diplomacy, before and following the simulation. Reflective essays are frequently used to internalize learnings from serious games as they enable participants to link their experiences in the simulation to the theories and approaches studied [5].

The simulation consisted of three rounds (see Fig 2 for game process). At the beginning of each round, every country introduced one of their draft articles for inclusion in the treaty, explained why it should be included, and answered queries

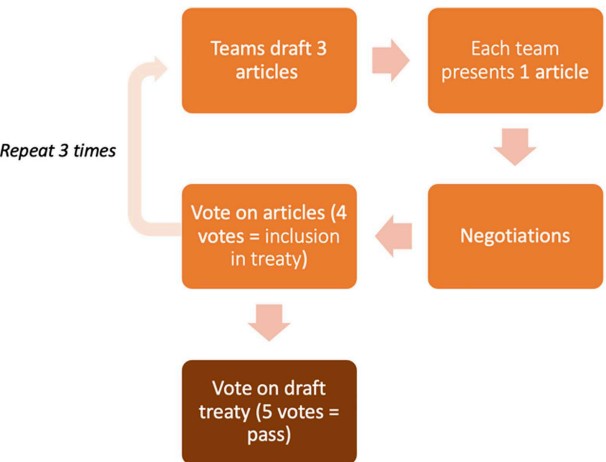

**Fig 2. Flow chart of game process.**

from the other country teams. Following the presentation of draft articles, all teams were provided access to the text of the proposed articles on a shared online document. There was then a period of negotiation where teams could try to solicit support for their proposed article, influence changes in others' articles, and/or find allies to combine similar articles. During negotiation periods, teams sought to influence each other through logical arguments and appealing to shared interests, as well as by trading PRPs in exchange for support. Following each round of negotiation a vote was conducted on revised articles to determine whether each article was to be included or not. Articles with support from at least four countries (i.e., four 'yes' votes) remained in the draft treaty, with sponsoring state(s) earning one PRP each. Articles which did not receive four 'yes' votes were discarded. This process was repeated three times.

The simulation also included 'chaos players.' Chaos players are used in serious games to inject a realistic element of the unexpected. During one round, the chaos player was a civil society organization concerned about data security, and expressing fear that global surveillance systems would threaten individual privacy. In another round, the chaos player was a multinational pharmaceutical company with access to PRPs that could be exchanged with countries for alignment with sectoral interests; for example in protecting intellectual property regulations and preventing development of generic medical products, such as vaccines. In the third round, the chaos player was a healthcare workers' union advocating for greater protections for healthcare workers. The purpose of introducing varying chaos players was to demonstrate the complex and divergent interests and influence of non-state actors in global health governance.

Once the three rounds of negotiations were complete, all the articles that had received at least four votes were combined into a full draft treaty. A final vote was then held across all country teams, on whether to adopt the pandemic treaty. Voting 'yes' cost two PRPs, but if the treaty was successfully adopted, all teams, regardless of if/how they voted, would earn three PRPs. This, again, was in effort to simulate the costs and benefits of collective action, as well as the potential for free riding. This learning was further buttressed by the structure wherein it was possible that the final draft included articles that teams may not have supported during earlier rounds of negotiations because only four 'yes' votes were initially required for article inclusion. Since five 'yes' votes were needed for the whole treaty to pass, teams were potentially faced with the choice of approving a treaty they mostly agreed to or rejecting the whole thing. In this instance, only three states (i.e., student teams) voted 'yes' to adopting the final draft of the treaty and so the pandemic treaty was not agreed to or adopted, and thus, all teams lost. Following the final vote on the treaty, students participated in a facilitated discussion and debrief. As multiple articles on serious games note, reflection and discussion after the simulation are crucial components to the learning success of a simulation [5].

## Assessment methodology

This was an exploratory, mixed method qualitative study including participant observation and thematic document analysis. Recruitment began on 2 December 2023 and ended on 15 January 2024. Students completed written informed consent forms, which were kept anonymous from the course instructor to ensure academic evaluations were independent from consent. Consent was sought for authors to act as participant observers and, after having received their final marks, for student reflective essays to be analyzed and quoted anonymously. Students could opt into either or both parts of the study (observation and/or thematic analysis) and were informed prior to the simulation that their consent to participate (or not) would have no bearing on their course grading. All students consented to participant observation and five of the six students consented that their assignments could be analysed and included.

As noted, the authors (the lecturer, teaching assistant, and guest lecturer) acted as participant observers, with one filling the role of GHO and the others participating in a country team. The authors took notes during the simulation and discussed observations following it in order to ensure reliability and identify biases (i.e., noting when the authors observations confirmed each other's or differed and why). In cases were observations differed; these differences were discussed until agreement was reached. The five student assignments were analyzed using a mixed deductive and inductive thematic analysis approach, similar to methods used by O'Dowd [32], to conduct thematic analysis of student assignments. Our first, deductive, set of themes corresponded to the learning objectives for the activity. In reading the assignments we also identified two further themes, which were refined through conversations among the co-authors, taking observations and notes into consideration (See Table 1 for list of themes). One author then conducted the thematic analysis of the assignments, sharing the analysis with the others who reviewed for consistency and accuracy. Suggested changes were discussed until all authors agreed. The authors also consulted their individual and group notes as participant observers, triangulating these with the findings of the student assignments.

## Results

### Learning objectives: Apply learning from past health crises

The analysis first asked if students demonstrated the ability to apply learnings from past health crises. Throughout the term students had engaged with several case studies of health crises through readings, discussions, and in-class activities. They were asked to draw on learnings from these cases when preparing for, engaging in, and reflecting on the Accord simulation. The application of this learning was evident throughout different aspects of the serious game; students referenced specific cases to strengthen their arguments and negotiating positions, in presenting their articles, and when explaining their learning in their reflective essays. Reflecting on the utility of the proposed pandemic treaty they and their colleagues had created, several students referred to different course case studies. For example, when discussing a proposed article on vaccine access one student wrote,

> "With COVAX, although the intention was to have equitable vaccine distribution, this was not the case because countries who said they would donate a certain number of vaccines ended up donating less. Without some sort of authority figure to issue penalties to ensure states comply to the terms of the pandemic treaty, it may render useless if nations do not play their part." (Student 2)

**Table 1. Thematic framework.**

| Learning Objective Themes | Additional themes |
|---|---|
| Apply learning from past health crises | Moral dilemmas |
| Actively engage with key concepts | Decolonizing learning |
| To experience global health cooperation strategies and challenges | |

Another student reflected on how the treaty would be actualized, noting past challenges in implementing global health responses.

> "One of our readings discussed how during the polio outbreak in Syria, due to the WHO's position of not interfering with state sovereignty, the WHO did not react and implement programs in rebel-controlled areas. This was briefly touched upon in Article 4.2 [of the proposed treaty] which calls for an allocation of resources and collaboration with humanitarian organizations in regions affected by conflict. Based on the insights from Syria, how would our article work when governments refuse to cooperate and what position would the GHO be expected to take considering the consequences of the WHO refusing to do anything against the wishes of the Syrian government?" (Student 1)

Reflections also moved beyond diagnosing the problems in past crisis response to conceptualizing solutions. A student reflected,

> "Thinking back on the Ebola outbreak in West Africa, the lack of infrastructure, health care workers, technologies and other resources needed for a strong health care system resulted in many cases and deaths. However, the drafted treaty acknowledges these, and I believe will help mitigate future pandemics by building on LICs capacity, therefore, alleviating the pandemic burden on a global scale." (Student 2).

Another student commented on an article their team had included around health delivery during conflict,

> "The inclusion of this article addresses the inaction that has occurred by leading global health actors respecting the authority of de jure governments in areas of contested sovereignty during previous health emergencies such as the 2013 Syrian polio. Incorporating global health partnerships allows for the treaty to be enacted in regions outside of member state jurisdiction, ensuring conflict-affected areas are not neglected during future pandemics." (Student 4)

Students applied learnings from case studies to illustrate key concepts, support their arguments, and construct solutions to overcome past challenges.

**Learning objectives: Actively engage with key concepts**

The course curriculum included a focus on key global health governance concepts including the securitization of health, intersectional and transnational feminism, political economy, and the commercial determinants of health. As health sciences graduate students, much of this content was new to the class. Analysis of assignments asked if there was evidence the game provided opportunities for students to actively engage with key concepts, finding that it did so. For example, one student wrote:

> "The chaos actors reminded me of our reading on the five vehicles of power that corporations and civil society use to impact global health decisions. In our case, the union was using the political environmental vehicle by lobbying, or corporate influence on decision-making through exclusive access to policy makers. I was reminded of the enormous power of non-state actors within global health governance." (Student 4)

Another student similarly reflected on "an example of how the treaty is shaped by politics. There is this claim and understanding of global health needs a united effort to overcoming global disease but there is a lack of initiative for accountability and responsibility." (Student 1)

Many students raised the concept of global health as a public good and discussed contradictions between this ideal and the self-interest of states. One student wrote, "The article proposed in the draft treaty did see vaccines and disease

eradication as a necessary public good.... [but] the limitations of the article accepted in the course pandemic treaty for Zecan were evident, as it offered a temporary three-year agreement with terms that primarily favored high-income countries like my nation, Swacor." (Student 5) Similarly, another student wrote, "There was a recognition of vaccines as global public goods but debate surrounding who pays for and has authority over them." (Student 3)

Other students applied feminist analysis to evaluate the treaty negotiation process.

"This component of the proposed pandemic treaty also highlighted course concepts which discussed how women, including those in working positions, are often differentially impacted from pandemic impacts. This is further highlighted in the course readings which discuss how women are seldom considered in the responses to global health emergencies, leading to long-term health and economic effects." (Student 5)

And

"[Course readings] demonstrated the ways women have been disproportionately impacted by disease outbreaks including the Ebola and Zika viruses, urging the need to address gender inequality and discrimination in emergency responses.... Despite this significant challenge, gender is only mentioned in Article 4.3 on healthcare workforce strengthening. Although this article mentions the need for gender equality in the healthcare workforce and its incorporation in related policies, it is in accordance with state laws and norms which leaves little room for change and progress in addressing the underlying issue." (Student 3)

Students applied feminist principals to both construct their articles, and also to critique the final treaty which they recognized as falling short on gender equity goals.

**Learning objectives: To experience global health cooperation strategies and challenges**

The simulations were designed to enable students to experience global health cooperation strategies and challenges in a fictional and highly constructed environment. Student reflections highlight which experiences particularly resonated with their learning. For example, many commented on time constraints: "During the two-and-a-half-hour negotiation process, I had to cut conversations short because I would need to engage in a conversation with a different party. It was overwhelming which made me think about what the actual negotiation process would be like with 188 more countries." (Student 1) The facilitator often had to urge students to complete negotiations and make decisions, as students had to manage both in-team discussions and conversations with other states, testing their collaborative working and communications skills. One student wrote,

"Engaging collaboratively with my partner and participating in negotiations with other teams provided me with valuable insights into the complex and intricate strategies that are critical in effective negotiation grounded in respect and mutual benefit. This challenged my thinking as I began to understand the difficulty and need to balance competing interests that exist between global health and international relations. (Student 3)

Numerous students commented on the challenge of reconciling domestic priorities with overall global health goals:

"This simulation showed me the difficulty of making connections and fostering collaboration when there are varying values. For example, during the last negotiation period, the three groups my team wished to discuss with were occupied deliberating intellectual property, while our focus was on surveillance. We tried to ask if we could speak to one representative from each country but the responses we got were, "We already have a surveillance system" or "This IP discussion is more important". (Student 2)

Others noted how different perspectives on equity concerns limited meaningful action.

"The difficulty of adopting gender transformative approaches due to the politics that often surround gendered health issues was apparent throughout the simulation... one country preferred gender-blind approaches to global health governance that refrained from highlighting gender inequalities within health emergencies... This disagreement around gender inequality during the simulation further illustrated the complexity of global health governance, as domestic politics transcend borders to influence global decisions." (Student 4)

Further reflection of the constraint of domestic policies on global processes from a country team with long standing international conflicts included, for example, that: "Representing Mepria in the pandemic treaty simulation, I began to recognize the complex relationships between local political climates and global health need... Additionally, as politics impacted our ability to form alliances or even negotiate with some nations, the collective global health need was often ignored, which was reflected in the draft treaty." (Student 3) Experiencing both practical and political constraints to negotiations, led to considerations of why the global health ideals expressed in international fora are so difficult to operationalize in practice.

### Additional outcomes: Experiencing moral dilemmas

The above learnings in turn led many students to experience and reflect on moral dilemmas arising from their efforts to reconcile their interests in negotiating an effective and equitable pandemic treaty and their material or political self-interests as diverse countries. For example, one student wrote:

"Discussions within my country team were geared towards maintaining our PRPs and monetary and trade success and keeping our citizens at ease during the pandemic and future pandemic treaty. It was challenging to understand global health and equity-based pandemic principles yet create selfish articles which would serve to benefit only high-income countries." (Student 5)

Many students noted they became distracted by wanting to earn or save PRPs: "Although the concept of points made it fun it was at times distracting because I found myself worrying about the points rather than what my country would do." (Student 1)

The role of Big Pharma, as a chaos player who could give away PRPs, also led to moral conflicts,

"I think one concept that stood out the most during the negotiation process was how much politics and corporations were involved. We all went into negotiations wanting the best for our nations and not the world as a whole. For example, my country was a large manufacturer of vaccines and we found ourselves trying to please the pharmaceutical company more than we did the LMICs because there was a possibility of us getting points." (Student 1)

And

It [taking PRPs from Big Pharma] was a morally conflicting experience; despite recognizing what was happening and knowing it was better to turn away, we became the victims of corporate power for financial incentive. It was surprising because I wanted to be a responsible public health professional, capable of refusing corporations and leading the many initiatives to halt their influence. Instead, I surrendered to secure the financial assistance my country needed. This experience revealed that even with the right moral principles, when confronted with adversity or, in this case, you lack the resources for your nation, circumstances may lead you to take actions contrary to your moral compass." (Student 2)

Other students noted that moral dilemmas were created by power structures that challenged ideals of democratic or fair global governance: "I assumed that, as everyone ultimately gets one vote, we would all have similar power during negotiations. However, with the lowest PRPs, we had limited resources to leverage for support. My teammate and I found ourselves compromising key values to build and maintain relationships." (Student 4) Similarly, a student wrote, "Because we were a low-income country with limited resources and only two PRPs, we were very cautious in selecting our article topics and parameters. We tried to articulate them to meet our dire country's needs but also phrased the articles in a way that would appeal to middle- and high-income countries to get their vote. I found this process to be quite difficult." (Student 2)

These comments, as well as discussions at the end of class revealed that students did not always behave as they had expected to during the simulation. Even though co-operation was essential to success, many noted with surprise how the concept of winning diverted the priorities they had set prior to the simulation. Participant observers also noted how focused on PRPs teams became, despite the game structure demanding co-operation, with this reflecting both a desire to 'win' the game and to do what was perceived as best for their country/team, which could be in conflict with global cooperation goals.

### Additional outcomes: Critical reflections on decolonizing health

Experiencing the moral dilemmas described above, as well as the power relationships inherent to negotiation processes led to critical reflections around decolonizing global health – a topic that was frequently discussed throughout the course, but which was not explicitly incorporated into instructions for this activity and assignment. One student wrote,

"The power relationships that were on display reminded me of our conversations around decolonization and this need to acknowledge the colonial context underpinning global health. Reflecting on the simulation, many articles focused on the conventional definition of global health as public health elsewhere. Unknowingly, a room full of professionals committed to decolonization centred discussions around a philanthropic and neocolonial understanding of global health governance that focused on elevating LMICs to the standards of HICs." (Student 3)

Students recognized that the treaty articles reinforced the colonial structures of global health.

"The treaty also does not address one of the major challenges discussed heavily in class which is decolonizing global health. This was predominantly evident in the technology transfer article where it is still very much seen as HICs being charitable to LMICs... I feel like this creates a feedback loop of LMICs still having to rely on these HICs because the capacity-building onus is placed on LMICs and the technology transfer support from HICs comes too late in the prevention stage. As discussed in our course, decolonizing global health emphasizes for equitable, inclusive and just public health that lacks supremacy." (Student 1)

Others noted a double standard in how responsibility was positioned in the treaty articles: "upon review of the articles, there are a lot of parameters regarding how the LMICs need to be supported but there is a lack of parameters regarding what the role of HICs is." (Student 1) Another student similarly wrote, "While the simulated treaty proposed during the activity effectively addressed numerous global health challenges, prioritizing low- and middle-income countries, I observed a notable oversight in capturing the influence of high-income countries in treaty negotiations." (Student 5)

Some students pointed out how specific issues were neglected due to a focus on "traditional/colonial priorities," writing "an element that was missing in the treaty was acknowledging Indigenous ways of knowing and ensuring the safety of Indigenous people." (Student 2) Another student noted,

"While Articles 4.2 and 4.3 mention traditional health providers and gender considerations, these concepts are relatively underdeveloped. Most articles centre around Western views of health and perpetuate the conspicuous invisibility

of women despite the acknowledgement of the role of women as caregivers. Addressing gender considerations and Indigenous health practices in separate articles could ensure these concepts are prioritized and facilitate dialogue around the historical, colonial, and political contexts that underlie inequities." (Student 4)

Similar to their experiences of unanticipated moral dilemmas, students were disappointed they had not taken a more decolonial approach to the treaty negotiations. However, they also identified how the structure of the negotiations and pre-existing power dynamics limited their ability to do so. One student commented during the end of class discussion that they now understood the decisions their home country (a LMIC) took in international fora better, as they had experienced the limitations imposed on LMICs. Despite awareness of calls to decolonize global health, students were limited in their ability to achieve this ideal by the very structures they would have liked to dismantle.

## Discussion

Serious games have been used in other disciplines to foster deep learning and critical thinking while building collaboration and communications skills. Our experience employing a simulation of a global health treaty negotiation demonstrated similar potential for global health courses. Indeed, simulations may be particularly apt tools for interdisciplinary domains such as global health, as they enable students to engage with various sets of knowledge at once and explore unfamiliar concepts. In this case, students who had limited prior knowledge of IR and related disciplines, demonstrated valuable learning, in particularly on the importance and influence of political economic interests in global health governance. Our experience adds further evidence to support Ho et al.'s suggestion that the use of such simulations be expanded beyond the fields of international relations and law [10].

Fabricatore et al. note that serious games may be particularly apt for learnings around how to manage wicked problems, as they engage players in collective activities that require the continuous interpretation of changing circumstances [33]. Global health is characterised by multiple wicked problems; enabling students to engage with those related to a pandemic response provided an opportunity to practice drawing on multiple perspectives and test various approaches. Students also drew on past case studies to develop ideas on how to overcome persistent challenges in global health, suggesting an advancement of their critical thinking skills beyond problem diagnosis to response construction. This ability to learn from past cases is a skill that has wide applicability for global health students as they move on to careers where they will be required to design or develop health interventions and/or policy.

Simulations are often used in both military and medical fields because they enable learners to test new approaches and ideas in a safe-to-fail environment [34]. In this case, all the teams did indeed 'fail' at the game as they could not achieve agreement on the treaty. Students were particularly dismayed by their inability to overcome self-interest to achieve a global pandemic treaty, and by the lack of equity and decolonizing considerations in the draft treaty. However, it is these 'failures' that provided opportunities for deep learning and critical reflection as students questioned what constrained their underlying motivations and actions. The tensions between the ideals often taught in global health classes – such as those related to gender equality – and the challenges of implementing them in global health processes were directly experienced. The simulation also required students to grapple with how such ideals are constrained by persistent power relationships including colonialism.

Khalid writes that simulations can prompt students to navigate and respect different cultural contexts, which in turn prepares them for handling sensitive issues [26]. We found that the fictional world of Zecan put students in roles they were uncomfortable and unfamiliar with, prompting them to view the challenge of pandemic response from alternative perspectives. This act of being placed in someone else's shoes generated both empathy towards those tasked with decision-making and humility around their own assumptions about what good global health governance entails.

This in turn fed into critical reflections around decolonizing global health. There is a growing literature on how to translate calls to decolonize global health into global health teaching. Banerjee et al. have challenged global health instructors

with the question, "are we training students to be "white saviours", irrespective of their ethnicity or country of origin, and how can global health education be reconfigured?" [35]. These questions, and broader movements to dismantle the White Industrial Saviour Complex and global hierarchies in education are complex, and by no means do we suggest serious games/simulations are the simple answer to these challenges. Nevertheless, we are encouraged that the critical reflections on this simulation included recognition of the ongoing harms of colonial structures and dominant power relationships. Critical pedagogies demonstrate how exposing contradictions can reduce the power of hegemonic ideas, and through this simulation students experienced the contradictions of a global health system that espouses equity but is buttressed by systems of colonization, including the power of multinational co-operations. Students noted the differences in how low-income and high-income countries were presented in articles, which topics were prioritized, and how, despite a one-member-one-vote system, some had much greater power than others. Though such learning can be achieved through other means, by actively experiencing these dynamics through the simulation, student learning and critical reflection was deepened and internalised through the moral conflicts which arose.

Classroom simulations are often evaluated using pre- and post- surveys to identify growth in learning around specific concepts and overall impressions. Yet, Starkey and Black note that, "The strength of simulations lies not primarily with improvements of the theoretical vocabulary and understanding, but with improvements of critical thinking and moral reasoning. The problem with that type of learning is that it is exceedingly difficult to capture in surveys" [34]. Noting this limitation to surveys, as well as our small class/sample size, we opted to use a combination of participant observation and thematic analysis of student assignments to explore whether and what learning outcomes had been achieved through the simulation. Still, we recognize our analysis is limited by the small class/sample size and qualitative methodology, which limits the generalizability of the results. Considering it was not possible to reach saturation within this sample, we note the need for further research and assessment on the use of serious games in global health teaching.

With these limitations in mind, we feel the data presented here indicates that the serious game Accord not only met the learning objectives of applying learning from past health crises, actively engaging with key concepts, and experiencing global health cooperation strategies and challenges, it also prompted deeper reflection on decolonizing global health and the moral dilemmas inherent in global health governance. While the 'failure' of the students to negotiate a successful treaty may suggest a negative outcome, it in fact provided a deep learning experience of the challenges to global health cooperation. The fact that the game so clearly mimicked what was occurring at the global level at the time – with the inability to reach consensus within the pandemic instrument negotiations in 2024 – further suggests such simulations provide an opportunity for students to engage in real world challenges while in the classroom. Though the duration of the course did not allow ongoing real-time comparisons, with the real Pandemic Treaty finally adopted at the World Health Assembly in May 2025, there is a further opportunity for students to compare their priorities and diplomacy approach and those of state and institutional actors. Students can reflect on their own single negotiation and consider how this deviates or aligns with other past global health negotiations. Furthermore, health policy is routinely changing, whether on a local or international scale, and the negotiation skills practiced by these students provided valuable insight into the process that can be applied to a multitude of environments. Similar games could be also developed around other global health processes, such as negotiations within development banks, or in response to specific events, including health crises.

## Conclusion

The Accord simulation effectively achieved its learning outcomes, in addition to fostering deep learning and critical thinking among the student participants. The in-class activity and discussions demanded students put their learning into practice, while the final essays prompted reflection. Equally important, all participants expressed that they enjoyed the activity and would recommend it to others. There appears to be great potential to apply serious games to global health learning, particularly where such activities foster critical reflection on, and tools for decolonizing global health knowledge and teaching.

## Acknowledgments

We are grateful to the students and colleagues who participated in the simulation and shared their insight with us, in particular, those who voluntarily participated including Haaris Tiwana, Alexandra Selinger, Lara Hollman, and Alice Murage. This work was funded in part by the Pandemics and Borders Project which receives funding from the Canadian Institutes of Health Research and New Frontiers in Research Fund administered by Social Sciences and Humanities Research Council of Canada (SSHRC).

## Author contributions

**Conceptualization:** Julia Smith, Ellie Gooderham.

**Formal analysis:** Julia Smith, Ellie Gooderham, Julianne Piper.

**Supervision:** Julia Smith.

**Writing – original draft:** Julia Smith.

**Writing – review & editing:** Ellie Gooderham, Julianne Piper.

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
