## [Decision Letter · Decision Letter 0]

23 Dec 2024

PGPH-D-24-01853

“I wanted to be a responsible public health professional”: applying serious games for deep learning in global health courses

Dear Dr. Smith,

Thank you for submitting your manuscript to PLOS Global Public Health. After careful consideration, we feel that it has merit but does not fully meet PLOS Global Public Health’s publication criteria as it currently stands. Therefore, we invite you to submit a revised version of the manuscript that addresses the points raised during the review process.

We look forward to receiving your revised manuscript.

Kind regards,

Julia Robinson

Executive Editor

Journal Requirements:

1. Please provide an Author Summary. This should appear in your manuscript between the Abstract (if applicable) and the Introduction, and should be 150–200 words long. The aim should be to make your findings accessible to a wide audience that includes both scientists and non-scientists. Sample summaries can be found on our website under Submission Guidelines:

https://journals.plos.org/globalpublichealth/s/submission-guidelines#loc-parts-of-a-submission.

2. In the online submission form, you indicated that Due to small sample size and qualitative nature of the data, de-identification is not possible. Therefore this data can not be made available. Game materials are available from the corresponding author on request.

a. In a public repository, 

b. Within the manuscript itself, or 

c. Uploaded as supplementary information.

Additional Editor Comments (if provided):

Reviewers' comments:

Reviewer's Responses to Questions

**Comments to the Author**

1. Does this manuscript meet PLOS Global Public Health’s publication criteria ? Is the manuscript technically sound, and do the data support the conclusions? The manuscript must describe methodologically and ethically rigorous research with conclusions that are appropriately drawn based on the data presented.

Reviewer #1: No

Reviewer #2: Yes

2. Has the statistical analysis been performed appropriately and rigorously?

Reviewer #1: N/A

Reviewer #2: N/A

3. Have the authors made all data underlying the findings in their manuscript fully available (please refer to the Data Availability Statement at the start of the manuscript PDF file)?

Reviewer #1: No

Reviewer #2: No

4. Is the manuscript presented in an intelligible fashion and written in standard English?

Reviewer #1: Yes

Reviewer #2: Yes

5. Review Comments to the Author

Reviewer #1: The concept presented in this manuscript is not only immportant but also holds the potential to significantly enhance global health education. The use of serious games, particularly in fostering interdisciplinary learning, is a promising avenue for advancing the field of global health pedagogy. However, due to several critical issues, I am compelled to recommend against publishing this manuscript in its current form.

1. Lack of comprehensive review of related experiences:

The manuscript aims to contribute to documenting serious games as a pedagogical approach in global health education. However, a comprehensive review of similar applications in global health and allied fields such as international relations, political science, and medical training is not just essential but also a potential area for further research and exploration. This review is crucial to substantiate the study's rationale and contextualise its contributions. As stated in the article, the study aims to "add to the evidence on serious games in global health pedagogy," yet there is no systematic search strategy or in-depth review to justify this claim and clarify the novelty of this study about existing literature.

2. Methodological concerns as a qualitative study:

While the study utilises participant observation and thematic analysis, it does not fully adhere to established guidelines for reporting qualitative research. As a result, several crucial elements remain ambiguous, notably:

Methodology: The study lacks a defined qualitative methodology, which is a crucial aspect that the authors need to address. With a clear methodological framework, the study's basis and approach to data analysis are adequately justified.

Coding and analysis process: The thematic analysis needs a more detailed explanation of the coding process, including steps for ensuring reliability, which are essential for robust qualitative findings.

Role of observers and bias management: Given the authors' observational role within the simulation, potential biases should be addressed. The manuscript would benefit from describing measures to maintain objectivity in data collection and analysis.

3. Small sample size:

The study is based on a very limited sample, and whether data saturation has been achieved remains to be determined. Although the study offers insights into learning outcomes from the simulation, a larger sample would be necessary to draw reliable and generalisable conclusions.

4. Absence of a comparison group:

A comparison between the serious game approach and more traditional teaching methods (e.g., lecture-based or practical sessions without gamification) would strengthen the study's claims about the unique effectiveness of serious games in this context. Such a comparison would help clarify the added value of gamification versus conventional approaches in achieving learning objectives.

In its current form, this manuscript does not yet serve as a rigorously peer-reviewed piece of evidence for the field. With further development, it could make a valuable contribution, yet at this stage, it may be better suited as a "Correspondence" or "Letter to the Editor" rather than a full original research article, given its lack of necessary components for a robust qualitative study.

Reviewer #2: Dear Editor,

Firstly, I would like to thank you to invite me to review this paper, especially because gaming, deep learning, AI and LLM’s application in global health are areas of great interest to me.

Overall, I found the paper to be a very original and interesting, with detailed explanation of the approach undertaken, the measurement of the results, the classroom setting and overall learning of the students – very rich food for thought for academicians involved in lesson planning and program design definitely, and even international conferences, which already provide an international and interdisciplinary setting.

I have some specific comments that would help in tightening this article, and are listed below.

Pg2 line 34. Add ‘were’ before ‘developed out of the tradition..’. There are few other typos/ missing words so a thorough read should do take care of this error.

Pg2 lines 46-54 are part of methodology and can be moved there.

Pg 3 lines 63-80. Overall a significant chunk of this information and other country examples can be moved to discussion section.

Pg3. Line 106-107 appears very abruptly after establishing the need of application of serious games/ simulation in global health teaching, without stating the width or depth of existing knowledge or any knowledge gap.

Pg 6 Line 127. Are there other games where fictional world settings have been used to simulate experiential learning, specially with curriculum of subjects like global health, which is very embedded in the current ‘global situation’ and its unique history and geography. A stronger rationale is required to justify the fictional world setting.

I feel that the introduction and methodology of the game Accord can be a part of the methods and materials, starting with description of the game, pre- and post-surveys etc. Methods section would not be the place where limitations or pluses of a methodology are discussed. Instead, these should be parked in the limitation section of the article. At the moment all this content, while coherent, is moving to and fro, and would benefit from a little bit of restructuring for better flow and removal of repetitions.

It would make for a catchier read if the results and students’ reflections from Pg 12 Lines 245 to Page 21 lines 496 are structured as per emerging themes based on the thematic analysis undertaken. Also, even though thematic analysis is mentioned, it would be valuable to elaborate on the emerging themes as well as the reason behind extracting these themes for the study or study objective. At the moment no themes have been discussed, which makes one wonder about the rationale of choosing to do thematic analysis, and the utility of this analysis.

Finally, a succinct summary of results, key learning outcomes and takeaways would nicely close the article, which can then be followed by conclusion. The takeaways should also include how the failures in the game have contributed to the learning curve of the students, aligned with or deviated from the desired learning outcomes, and how can similar models be developed and used.

In the end, coming towards the title of the article, the quote ‘I wanted to be a responsible public health professional’ somehow does not seem to effectively convey what the article is about. I would suggest titles indicating simulation, or serious games in action, or using them in wicked global health challenges, to develop critical thinking and cooperation and collaboration skills.

6. PLOS authors have the option to publish the peer review history of their article (what does this mean? ). If published, this will include your full peer review and any attached files.

**Do you want your identity to be public for this peer review?** For information about this choice, including consent withdrawal, please see our Privacy Policy .

Reviewer #1: **Yes: ** Reza Majdzadeh

Reviewer #2: No

---

## [Decision Letter · Decision Letter 1]

12 Jun 2025

PGPH-D-24-01853R1

Learning from Failure: Simulating pandemic agreement negotiations in a global health classroom

Dear Dr. Smith,

Thank you for submitting your manuscript to PLOS Global Public Health. After careful consideration, we feel that it has merit but does not fully meet PLOS Global Public Health’s publication criteria as it currently stands. Therefore, we invite you to submit a revised version of the manuscript that addresses the points raised during the review process.

The revisions made have strengthened the piece, however following further review have confirmed the need for some further minor revisions. We believe the two reviewers' comments have some overlap and so some comments may be addressed simultaneously. These edits will help place the work in an important broader context. 

We look forward to receiving your revised manuscript.

Kind regards,

Alexandra L. Phelan

Academic Editor

Journal Requirements:

Additional Editor Comments (if provided):

Thank-you for your manuscript which expands the literature on simulations for global health and global health law-making. We were very pleased to review the revisions made, however reviewers had some follow up minor revisions that are necessary to strengthen your piece. With these minor revisions addressed, I would be very happy to accept your piece for publication. Please find the reviewer comments below.

Reviewer 3

It has been inspiring to learn about this valuable student activity for engaging with "serious games" in global health. The student comments throughout the analysis reflect the thoughtful engagement of students in this pedagogical exercise and their support in evaluating it in the academic literature.

Prior to publication, I would recommend the following additional points be addressed in the:

* Introduction: highlighting the wider range of Model World Health Assemblies that have been launched in recent years, bridging Model UN activities and this mock Pandemic Agreement negotiation. https://www.ghspjournal.org/content/5/1/164.short

* Methods: discussing further how class discussions drew from the global health diplomacy literature to prepare them for Accord exercise.

* Discussion: considering how -- now that the pandemic agreement is complete -- students could engage with a past policy reforms, comparing their negotiations with the actual diplomacy in the World Health Assembly.

Reviewer 4

Summary of Article

This article describes the value of simulations – specifically a singular simulation on pandemic treaty negotiations with fictionalized countries and a point system – for global health pedagogy. The benefits it highlights are opportunities for deep learning, to integrate elements of the curriculum including different perspectives and approaches, considerations of moral questions in global health, and for students to experience the practical challenges of global health cooperation while developing interpersonal skills.

Overall Impressions

The revisions the authors made to the article are significant improvements, particularly in the description of the qualitative methodology, and the explicit and detailed recognition of the study’s limitations on account of its small sample size and exploratory, mixed method qualitative approach. The discussion of these limitations is useful, though may not be sufficient to overcome the article’s primary challenge, which remains that it is largely a reflection on the pedagogical benefits of the particular exercise (including discussion of the value of the points system – not necessarily a feature of simulated negotiations on international health treaties more broadly), as opposed to a discussion of the benefits of simulated negotiations generally to global health and global health law negotiations. Student reflections take up a considerable part of the article.

Additional Research and Analysis Recommended

The authors recognize the lack of literature on simulation exercises in global health law, and helpfully include revisions that refer to simulations in other fields including political science, medicine, nursing and pharmacy, and on One Health. The example of a meta-analysis of simulation exercises in medical education is welcome. Though this article would benefit from discussion of literature on the problem-based learning benefits of simulation exercises in other international lawmaking contexts and reasoning by analogy – for example, there is ample literature on the use of Model UN simulation exercises. Connections to this literature would support the authors’ assessment of the pandemic agreement simulation. As global health law is developed through UN agencies and organs, the drawing of that analogy would be acceptable.

Additional primary research could bolster the findings of this article, potentially including a survey of courses within the field of global health law that include simulation exercises (mindful of any ethical / privacy considerations there).

The article would be strengthened by a broader discussion of why these simulation exercises are specifically useful for the teaching of global health as a field. The authors’ analysis on the alignment of these exercises with goals of decolonizing global health is useful, as is the analysis identifying global health challenges as wicked problems. More could be provided on the benefit of these exercises in light of the revitalization of global health lawmaking to undertake the reforms the article describes (the pandemic agreement and the IHR reforms).

Reviewers' comments:

Reviewer's Responses to Questions

**Comments to the Author**

1. If the authors have adequately addressed your comments raised in a previous round of review and you feel that this manuscript is now acceptable for publication, you may indicate that here to bypass the “Comments to the Author” section, enter your conflict of interest statement in the “Confidential to Editor” section, and submit your "Accept" recommendation.

Reviewer #3: All comments have been addressed

Reviewer #4: (No Response)

2. Does this manuscript meet PLOS Global Public Health’s publication criteria ? Is the manuscript technically sound, and do the data support the conclusions? The manuscript must describe methodologically and ethically rigorous research with conclusions that are appropriately drawn based on the data presented.

Reviewer #3: Yes

Reviewer #4: Partly

3. Has the statistical analysis been performed appropriately and rigorously?

Reviewer #3: N/A

Reviewer #4: N/A

4. Have the authors made all data underlying the findings in their manuscript fully available (please refer to the Data Availability Statement at the start of the manuscript PDF file)?

Reviewer #3: Yes

Reviewer #4: Yes

5. Is the manuscript presented in an intelligible fashion and written in standard English?

Reviewer #3: Yes

Reviewer #4: Yes

6. Review Comments to the Author

Reviewer #3: It has been inspiring to learn about this valuable student activity for engaging with "serious games" in global health. The student comments throughout the analysis reflect the thoughtful engagement of students in this pedagogical exercise and their support in evaluating it in the academic literature.

Prior to publication, I would recommend the following additional points be addressed in the:

* Introduction: highlighting the wider range of Model World Health Assemblies that have been launched in recent years, bridging Model UN activities and this mock Pandemic Agreement negotiation. https://www.ghspjournal.org/content/5/1/164.short

* Methods: discussing further how class discussions drew from the global health diplomacy literature to prepare them for Accord exercise.

* Discussion: considering how -- now that the pandemic agreement is complete -- students could engage with a past policy reforms, comparing their negotiations with the actual diplomacy in the World Health Assembly.

Reviewer #4: Summary of Article

This article describes the value of simulations – specifically a singular simulation on pandemic treaty negotiations with fictionalized countries and a point system – for global health pedagogy. The benefits it highlights are opportunities for deep learning, to integrate elements of the curriculum including different perspectives and approaches, considerations of moral questions in global health, and for students to experience the practical challenges of global health cooperation while developing interpersonal skills.

Overall Impressions

The revisions the authors made to the article are significant improvements, particularly in the description of the qualitative methodology, and the explicit and detailed recognition of the study’s limitations on account of its small sample size and exploratory, mixed method qualitative approach. The discussion of these limitations is useful, though may not be sufficient to overcome the article’s primary challenge, which remains that it is largely a reflection on the pedagogical benefits of the particular exercise (including discussion of the value of the points system – not necessarily a feature of simulated negotiations on international health treaties more broadly), as opposed to a discussion of the benefits of simulated negotiations generally to global health and global health law negotiations. Student reflections take up a considerable part of the article.

Additional Research and Analysis Recommended

The authors recognize the lack of literature on simulation exercises in global health law, and helpfully include revisions that refer to simulations in other fields including political science, medicine, nursing and pharmacy, and on One Health. The example of a meta-analysis of simulation exercises in medical education is welcome. Though this article would benefit from discussion of literature on the problem-based learning benefits of simulation exercises in other international lawmaking contexts and reasoning by analogy – for example, there is ample literature on the use of Model UN simulation exercises. Connections to this literature would support the authors’ assessment of the pandemic agreement simulation. As global health law is developed through UN agencies and organs, the drawing of that analogy would be acceptable.

Additional primary research could bolster the findings of this article, potentially including a survey of courses within the field of global health law that include simulation exercises (mindful of any ethical / privacy considerations there).

The article would be strengthened by a broader discussion of why these simulation exercises are specifically useful for the teaching of global health as a field. The authors’ analysis on the alignment of these exercises with goals of decolonizing global health is useful, as is the analysis identifying global health challenges as wicked problems. More could be provided on the benefit of these exercises in light of the revitalization of global health lawmaking to undertake the reforms the article describes (the pandemic agreement and the IHR reforms).

7. PLOS authors have the option to publish the peer review history of their article (what does this mean? ). If published, this will include your full peer review and any attached files.

**Do you want your identity to be public for this peer review?** For information about this choice, including consent withdrawal, please see our Privacy Policy .

Reviewer #3: No

Reviewer #4: No

---

## [Editor Report · Decision Letter 2]

24 Jul 2025

Learning from Failure: Simulating pandemic agreement negotiations in a global health classroom

PGPH-D-24-01853R2

Dear Dr Smith,

We are pleased to inform you that your manuscript 'Learning from Failure: Simulating pandemic agreement negotiations in a global health classroom' has been provisionally accepted for publication in PLOS Global Public Health.

Best regards,

Alexandra L. Phelan

Academic Editor

Thank-you for your thoughtful and considered edits in the most recent revision. We are delighted to accept your manuscript for publication.